# Willingness to Pay for a Dating App: Psychological Correlates

**DOI:** 10.3390/ijerph20032101

**Published:** 2023-01-24

**Authors:** Lucien Rochat, Elena Orita, Emilien Jeannot, Sophia Achab, Yasser Khazaal

**Affiliations:** 1UniDistance Suisse, Faculty of Psychology, Schinerstrasse 18, 3900 Brig, Switzerland; 2ReConnecte Treatment Centre, Addiction Division, Department of Psychiatry, Geneva University Hospitals, Rue du Grand-Pré 70C, 1202 Geneva, Switzerland; 3Faculty of Biology and Medicine, Lausanne University, Rue du Bugnon 21, 1005 Lausanne, Switzerland; 4Addiction Medicine, Department of Psychiatry, Lausanne University Hospital and Lausanne University, Bugnon 23 a, 1011 Lausanne, Switzerland; 5Institute of Global Health, Faculty of Medicine, Chemin de Mines 9, 1202 Geneva, Switzerland; 6Clinical and Sociological Research Unit, Department of Psychiatry, Faculty of Medicine, University of Geneva, 1211 Geneva, Switzerland; 7Department of Psychiatry and Addictology, University of Montreal, Montreal, QC H3T 1J4, Canada

**Keywords:** Tinder, app, cybersex, dating, impulsivity, motives, gender, subscription

## Abstract

The smartphone dating app, Tinder, has become hugely popular in recent years. Although most people use a free version of the app, some pay for an augmented version to improve their experience. However, there is little evidence of the association between the willingness to pay for a dating app such as Tinder and users’ psychological characteristics. This study thus aims to compare Tinder paying versus non-paying users in terms of their pattern of use, excessive use of Tinder, motives for using Tinder, impulsivity traits, depressive mood, and sociodemographic variables, as well as to examine which variables best predict group membership. A total of 1159 Tinder users participated in an online survey. Group comparisons indicated that payers were more frequently male, reported greater motives for using Tinder than non-payers, and differed in their pattern of use compared with non-payers. Impulsivity traits did not significantly differ between the two groups. Being male and reporting greater motives for Tinder use significantly predicted being a payer. These findings provide insights into the processes that stimulate users’ greater consumption of online dating apps, such as reinforcement mechanisms and reward sensitivity.

## 1. Introduction

Tinder is a mobile dating application which has become hugely popular in recent years, with more than 1.4 billion swipes per day in the US and 50 million users around the world [1]. Among these users, about 6.6 million people are paying Tinder subscribers worldwide, which made this app one of the highest grossing non-game apps in terms of overall revenue in 2021 [2]. Tinder direct revenue amounted to USD 1.4 billion as of 2020, an increase of 18% from previous years [3,4]. The popularity of Tinder is heavily influenced by its design, which facilitates access, as no detailed profile or questionnaire is required to create an account, and it can also be accessed directly with a Facebook account. Furthermore, as an app available on smartphones only, it has key unique characteristics, such as enhanced portability, constant access, and geolocalization capacity [5]. Tinder’s main goal is to help people find a potential romantic and/or sexual partner, filtered by preferences such as gender, age, and location. With a simple finger “swipe," users can either like or reject the profiles that are proposed in their geographical vicinity. If two people have “liked” each other, they “match” and can therefore get involved in online and/or offline contact [6]. 

More attractive but paying versions of the app, such as Tinder Plus, Gold, or Platinum, have been developed since 2015. In contrast to the free version of Tinder, they grant users an unlimited number of likes per day (versus only 50 every 12 h in the free version), the possibility of changing geolocalization (“Passport”), being one of the top profiles in one’s area for 30 min (“Boost”), rewinding unliked profiles (“Rewind”), letting a potential match know they stand out (five “Super likes” versus only one per day in the free version), seeing who likes the participant before deciding whether to like them or not (“Likes you”), highlighting participants’ most swipe-worthy potential matches (“Top picks”), or attaching a message to a Super like [7], to name the main features. Some of these upgrades (e.g., Boost or more Super likes per day) can also be obtained by purchasing immediate upgrades on the go (“à la carte” purchase).

Although several studies have underlined that dating application can be beneficial to many users, others have stressed that such apps can be associated with addictive behavior, such as loss of control, conflict, tolerance, withdrawal, salience, or coping [8], and increased risky [9,10] or non-consensual sex [11]. From this perspective, several studies have examined the psychological characteristic associated with Tinder use and misuse. More specifically, impulsivity traits, motives for use, attachment style, sexual desire, and self-esteem have been associated with problematic Tinder use in a large sample of Tinder users. It was more specifically found that participants with the greatest level of problematic Tinder use showed greater enhancement, coping, and social motives for using Tinder, as well as a greater level of impulsivity traits linked to sensation seeking (which reflects pursuing exciting new experiences or risky activities) and urgency-related traits (which reflects acting rashly while in an intense positive or negative affective state) [12].

However, there is little evidence of the association between the willingness to pay for dating apps and users’ psychological characteristics. Regarding app use at a broader level, higher intention to use an app is linked to a higher probability of paying for it [13]. Thus, in consideration of the enormous success of Tinder, examining the psychological correlates of the willingness to pay for upgrades of this app, and investigating the potential association between Tinder use and problematic use remains an important question to investigate.

The aim of this study was first to compare Tinder paying versus non-paying users’ in sociodemographic variables, problematic Tinder use, Tinder-use patterns, motives for using Tinder, impulsivity traits, as well as depressive mood. Second, we examined which factors among sociodemographic variables, motives for using Tinder, impulsivity traits, or depressive mood best predicted group status. We hypothesized that being a man, greater motives for using Tinder, and impulsivity traits (especially urgency-related traits and sensation seeking, which have previously been associated with greater problematic use of Tinder), would be significantly associated with being a paying Tinder user [12]. Indeed, as these two dimensions of impulsivity have been associated with poor inhibitory control and difficulty in overcoming immediate gratification [14], we might expect that the possibility of immediately improving one’s experience on Tinder by paying for an extra service would be particularly appealing to impulsive users. As several studies have showed that Tinder use has been associated with lower well-being (e.g., depressive mood, anxiety) [12,15] and that depressive mood frequently co-occurs with impulsivity, especially the urgency trait [16], depressive mood has been used as a control variable in the current study. In addition, the relationship between motives for using Tinder and group membership could be fueled by impulsivity traits. Consequently, we hypothesized that impulsivity traits moderated the association between motives for using Tinder and group status; that is, a greater impulsivity trait should increase the strength of the association between motives for using Tinder and paying for the app.

## 2. Materials and Methods

This study used secondary data from a large study on the psychological determinants of Tinder use [12]. Participants were recruited in 2018 on various social networking sites (such as Facebook and Instagram) where a link to the survey was posted. They were all English-speaking Tinder users and were at least 18 years old. After removing 538 participants (322 with too many missing or incomplete data and 216 who reported a non-heterosexual orientation), the final sample consisted of 1159 subjects (46.59% women). The average age of the participants was 30.02 (standard deviation = 9.19, min = 18 and max=74). Of the participants, 66% were in a couple or married. Among the participants, 94 (8.11% of the final sample) reported paying for a subscription and/or paying for regular offers (e.g., microtransactions offering to buy Super likes or to boost visibility) on Tinder.

### 2.1. Measures

#### 2.1.1. Problematic Tinder Use Scale (PTUS)

The PTUS is a 6-item self-report measure assessing the six features of addiction as defined in the component model of addiction: salience, tolerance, mood modification, relapse, withdrawal, and conflict. Items are scored on a 5-point Likert scale (1 = never; 5 = always). Higher scores indicate greater problematic use. The PTUS showed good factor structure and moderate internal consistency [8]. In the current study, the internal consistency (Cronbach’s alpha) was 0.84.

#### 2.1.2. Tinder Use Patterns

This self-report questionnaire assesses the number of Tinder-initiated online and offline contacts in the preceding 6 months (1 = 0 people; 8 = more than 50 people); looking for committed romantic partners (1 = not true at all; 7 = absolutely true); looking for sexual partners (1 = not true at all; 7 = absolutely true); the number of current matches indicated on the app; satisfaction with Tinder (1 = not at all; 5 = entirely yes); and time since starting to use Tinder (1 = less than 3 months; 5 = more than 2 years).

#### 2.1.3. Short UPPS-P Impulsivity Behavior Scale (S-UPPS-P)

The S-UPPS-P scale is a 20-item self-report questionnaire examining five factors of impulsivity: positive urgency (e.g., “When I’m happy, I often can’t stop myself from going overboard”), negative urgency (e.g., “When I feel rejected, I often say things that I later regret”), (lack of) perseverance (e.g., “I am a person who always gets the job done”), (lack of) premeditation (e.g., “I usually make up my mind through careful reasoning”), and sensation seeking (e.g., “I like taking risks”). Items are scored on a 4-point scale (1 = I agree strongly; 4 = I disagree strongly). Higher scores suggest greater impulsivity. This questionnaire showed good internal consistency, test–retest stability, and predictive validity [17]. In the current study, the internal consistency (Cronbach’s alpha) was 0.82 (negative urgency), 0.73 (positive urgency), 0.79 (lack of premeditation), 0.80 (lack of perseverance), and 0.81 (sensation seeking).

#### 2.1.4. Cybersex Motives Questionnaire (CMQ)

The CMQ is a 14-item self-report questionnaire examining three motives for cybersex, including enhancement (i.e., to increase positive emotions, e.g., “to be entertained”), coping (i.e., to decrease negative affect, e.g., “to forget my problems”), and social (i.e., to increase social affiliation, e.g., “because I need to socialize with others”). Items are scored on a 5-point Likert scale (1 = never; 5 = always or almost always), with higher scores suggesting greater endorsement of the motive. This measure showed good factor structure and internal consistency [18]. For the purposes of the current study, scale instructions were modified to address Tinder use only. In the current study, the internal consistency (Cronbach’s alpha) was 0.84 (enhancement), 0.84 (coping), and 0.75 (social).

#### 2.1.5. Short Happiness and Depression Scale (SDHS)

The SDHS is a 6-item scale examining happiness (e.g., “I feel happy”) or depression (e.g., “I feel dissatisfied with my life”). Scale items are rated on a 4-point Likert scale (1 = never;4 = often). Happiness-related items were reversed so that higher scores reflected depressive mood. The SDHS has appropriate internal consistency, test–retest reliability, as well as fair convergent and discriminant validity [19]. In the current study, the internal consistency (Cronbach’s alpha) was 0.86.

### 2.2. Ethics

The study is a secondary analysis of a previous study on Tinder use [12] and is part of a larger project on cybersex. The original study [12] focused on the various psychological profiles of Tinder users, whereas the current study specifically aims to compare payers vs. non-payers and determine which factors account for group membership. The same data set was thus used in the current study, with the inclusion of an additional variable of interest, namely, payers vs. non-payers. The protocol was approved by the Ethical Committee of the Geneva University Hospital and was carried out in accordance with the Declaration of Helsinki. All participants received online information about the study before providing informed consent online and completing the questionnaires anonymously via SurveyMonkey links. The survey responses were sent over a secure, SSL-encrypted connection. 

### 2.3. Data Analyses

Mann–Whitney U comparison test (*Z*) was used to compare Tinder payers versus non-payers on continuous and ordinal variables. Non-parametric testwas performed because of the large difference in sample size between the two groups and the associated risk for assumption violation of parametric tests. In addition, Pearson’s chi-squared test was used to examine the association between categorical data (gender and relationship status) and participants’ status (payers versus non-payers). Second, a binary regression analysis was performed to examine the relationships between Tinder status (payers versus non-payers) and impulsivity traits, depressive mood, demographic data, and motives for using Tinder. As part of this analysis, we also examined the interaction between motives for using Tinder and impulsivity traits. All continuous variables were mean centered before performing the binary regression analysis, and all significant interactions were probed using the Excel spreadsheet provided here: http://www.jeremydawson.co.uk/slopes.htm, accessed on 11 May 2021. A *p* value of 0.001 was chosen as a cut-off for statistical significance to guard against Type I errors given the large number of analyses. All analyses were two tailed.

## 3. Results

### 3.1. Group Comparisons

Group comparisons (Table 1) indicated that men were significantly overrepresented in the group of Tinder payers. In addition, Tinder payers showed a significantly greater number of online and offline contacts during the past six months, greater satisfaction with Tinder, as well as looking more for both committed or sexual partners than non-payers. Age, positive urgency, and sensation seeking were higher in the payers than in the non-payers, although these results failed to reach statistical significance. No other comparisons reached statistical significance. Notably, after removing participants who both subscribed and paid for regular offers (N = 10), additional analyses comparing participants who paid for a subscription (N = 49) with those who paid for timely transaction offers only (N = 35) showed no statistical differences in all these variables (see Appendix A).

### 3.2. Binary Regression Analysis

A binary logistic regression was performed on the dichotomized variable, Tinder paying versus non-paying users with independent variables, including age, gender, marital status, cybersex motives (CMQ), impulsivity traits (S-UPPS-P), and depressive mood (SDHS), to predict group membership. To avoid any multicollinearity-related issues in the regression analysis associated with the strong correlations between the three factors of the CMQ (*r* range: 0.62 to 0.70; all *p*s < 0.0001), these three motives (social, coping, enhancement) were merged into a general cybersex motives factor. 

The final analyses indicated that the full model containing age, gender, marital status, impulsivity traits, cybersex motives, and depressive mood was statistically significant compared with the model with the constant only, χ^2^ (12) = 94.423, *p* < 0.001. The model, as a whole, explained between 8% (Cox and Snell) and 19% (Nagelkerke R squared) of the variance in group status and correctly classified 92.2% of the cases. The regression output (Table 2) showed that only gender and cybersex motives made a unique, statistically significant contribution to the model. In other words, more prominent cybersex motives and being male were both significantly associated with an increased likelihood of belonging to the Tinder paying users group. However, the interaction terms motives by impulsivity traits did not reach statistical significance.

## 4. Discussion

The objectives of this study were to compare Tinder paying versus non-paying users on various sociodemographic variables, Tinder-use patterns, problematic Tinder use, motives for using Tinder, impulsivity traits, and depressive mood, and to examine which variables best predicted group status. The main results showed that Tinder paying users were more frequently men, reported more severe problematic Tinder use, more online and offline contacts, as well as an increased intention to meet committed and sexual partners than non-payers. They also reported greater motives for using Tinder (coping, social, enhancement motives) and greater satisfaction with Tinder use. In the binary regression analysis, only being male and motives for using Tinder significantly predicted group status, thereby corroborating our hypotheses. However, in contrast with our expectations, there was no significant effect of impulsivity traits nor was there any significant interaction between motives for using Tinder and impulsivity traits in predicting group membership.

First, in accordance with our hypothesis, the results corroborate previous studies stressing that motives for using Tinder, such as physical (e.g., need for sexual pleasure), social (e.g., finding a romantic partner or friendships), as well as psychological (e.g., needs related to self-worth, such as validating the sexual attractiveness of one’s own appearance or restoring self-esteem) gratification stimulate users’ consumption of online dating applications [6,20,21,22,23,24]. In this context, the benefits of using a paying version of Tinder, such as having an unlimited number of likes, more Super likes, or being one of the top profiles in one’s area for 30 min, perfectly suits users’ motives and likely reinforces the expected benefits of such motives. Consequently, payers develop a greater intention to meet committed or sexual partners, effectively increase their number of online and offline contacts, and report more problematic use of the app. The greater reported satisfaction with use of the app in payers versus non-payers could also contribute to keeping users captive within the app through a process of positive reinforcement.

Second, the overrepresentation of men in the paying group is in line with data stressing gender differences in the perceived amount of attention received by dating apps or site users [25]. Indeed, men reported feeling as if they did not receive enough messages on dating apps or sites, whereas women tended to report that they were sent too many messages on dating apps or sites. In addition, women were more prone than men to believe that dating sites and apps are not a safe way to meet someone [25]. The overrepresentation of males in the payers versus the non-payers group could be associated with higher reward sensitivity or difficulty in delaying gratification in men. The literature has indeed underlined sex differences in reward processing: men showed higher reward sensitivity and sensation seeking in self-reports, and were more prone than women to take risks in laboratory tasks [26]. Paying to improve and increase the possibility of quickly meeting a partner on Tinder in one’s vicinity, and to accumulate dating partners is in line with greater reward sensitivity in men. 

This study is not without limitations. First, as the sample is self-selected, the generalizability of the results to the entire population of Tinder users is limited [27]. Moreover, the study relies exclusively on self-reports, an element that has been associated with various biases, including social desirability. Finally, the number of participants with too many missing or incomplete data (19% of the initial sample) who have been removed from the final sample may affect the representativeness of the current sample.

## 5. Conclusions and Further Perspectives

The current study adds to the field by increasing the body of knowledge on the psychosocial determinants of dating apps use and their misuse. These results can have clinical utility, considering the need for a personalized and multi-dimensional approach to the assessment and psychotherapy of social network misuse [28], as well as specifically those behaviors related to sexual behaviors [29]. Further studies should more specifically examine the amount invested by payers, the types of features participants are more willing to pay for, and which features are judged more efficient with respect to successful or excessive Tinder use. Further studies should also more specifically probe the interaction between impulsivity traits (e.g., positive urgency or sensation seeking) and motives in large subgroups of payers. Indeed, we anticipate that the interaction motive of positive urgency, which did not reach statistical significance in the current study, will be specifically stronger for those who use timely transaction offers as opposed to monthly subscriptions. Indeed, the former may show more difficulties in preventing themselves from paying to improve their experience on Tinder on the spur of the moment, which may maintain or increase their positive emotions. However, the small number of paying Tinder users, especially those who paid for timely transaction offers in our sample, prevented us from conducting such an analysis. Finally, further studies should determine whether problematic Tinder use in paying users actually leads to more long-term, significant negative consequences at a personal, medical, and social level for non-payers.

## Figures and Tables

**Table 1 ijerph-20-02101-t001:** Descriptive statistics and group comparisons between Tinder payers vs. non-payers.

	Payers (N = 94)	Non-payers (N = 1065)		
Variables	M (SD)	M (SD)	Z/Chi-Squared	*p*-Value
*Sociodemographics*				
Age	31.70 (9.31)	29.87 (9.17)	−2.13	0.03
Sex (% male vs. female)	80/20	51/49	28.61	<0.001
Relationship status (% single vs. in couple or married)	29/71	34/66	1.18	0.28
*Tinder-related*				
PTUS	2.68 (0.90)	1.85 (0.65)	−6.46	<0.001
Online contacts (med = 4 vs. 3)	4.27 (1.75)	3.26 (1.85)	−4.99	<0.001
Offline contacts (med = 3 vs. 1)	3.14 (1.52)	1.91 (1.21)	−8.34	<0.001
Committed partners (med= 4 vs. 3)	4.35 (1.69)	3.13 (1.93)	−6.01	<0.001
Sexual partners (med = 5 vs. 3)	4.70 (1.75)	3.32 (2.02)	−6.31	<0.001
Current matches	39.03 (106.03)	35.94 (119.84)	−1.96	0.05
Satisfaction with Tinder (med = 3 vs. 2)	2.86 (0.65)	2.34 (0.81)	−6.00	<0.001
Time since using Tinder (med = 2 vs. 2)	2.45 (1.29)	2.54 (1.52)	−0.15	0.87
*Other questionnaires*				
UPPS_Negative urgency	2.64 (0.69)	2.61 (0.69)	−0.82	0.41
UPPS_Positive urgency	2.76 (0.64)	2.63 (0.57)	−2.20	0.03
UPPS_Lack of premeditation	1.90 (0.55)	1.85 (0.52)	−0.83	0.41
UPPS_Lack of perseverance	1.91 (0.56)	1.95 (0.55)	−0.68	0.49
UPPS_Sensation seeking	2.87 (0.69)	2.73 (0.61)	−2.38	0.02
SDHS_Depressive mood	2.27 (0.63)	2.18 (0.66)	−1.67	0.09
CMQ_Enhancement	3.09 (0.80)	2.62 (0.79)	−5.54	<0.001
CMQ_Coping	2.88 (1.02)	2.12 (0.94)	−6.69	<0.001
CMQ_Social	3.25 (0.86)	2.62 (0.93)	−6.22	<0.001

Note. PTUS: Problematic Tinder Use Scale; UPPS: Urgency–Premeditation–Perseverance–Sensation-seeking Impulsivity Behavior scale. SDHS: Short Happiness and Depression Scale; CMQ: Cybersex Motives Questionnaire.

**Table 2 ijerph-20-02101-t002:** Binary logistic regression analysis on 1159 Tinder payers vs. non-payers.

Variables	β	SE_β_	χ^2^ (Wald’s)	*p*	e^β^ (Odds Ratio)	95% CI for e^β^	
						Lower	Upper
Age	0.02	0.01	3.14	0.08	1.02	0.99	1.05
Sex	1.25	0.28	20.26	<0.001	3.49	2.02	6.00
Marital status	−0.29	0.26	1.19	0.28	0.75	0.45	1.26
CMQ_ Cybersex motives	1.10	0.19	34.02	<0.001	3.00	2.08	4.35
UPPS_Negative urgency	−0.43	0.22	3.63	0.06	0.65	0.42	1.01
UPPS_Positive urgency	−0.16	0.30	0.29	0.59	0.85	0.47	1.54
UPPS_Lack of premeditation	0.33	0.26	1.60	0.21	1.39	0.83	2.32
UPPS_Lack of perseverance	−0.24	0.26	0.83	0.36	0.79	0.48	1.31
UPPS_Sensation-seeking	−0.01	0.25	0.00	0.98	0.99	0.61	1.62
SDHS_Depressive mood	0.25	0.20	1.65	0.20	1.29	0.88	1.90
Cybersex motives x Positive urgency	0.73	0.31	5.71	0.02	2.08	1.14	3.78
Cybersex motives x Sensation seeking	−0.21	0.30	0.48	0.49	0.81	0.54	1.46
Constant	−3.60	0.27	173.24	<0.001	0.03	-	-

Note. CMQ: Cybersex Motives Questionnaire; UPPS: Urgency–Premeditation–Perseverance–Sensation-seeking Impulsivity Behavior scale. SDHS: Short Happiness and Depression Scale; Sex was coded 1 (male) vs. 0 (female). Marital status was coded 1 (in couple/married) vs. 0 (single).

## Data Availability

Data will be made available by the authors upon reasonable request.

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
