# Peer review of "Willingness to Pay for a Dating App: Psychological Correlates"

_ijerph, 2023, doi:10.3390/ijerph20032101_

Round 1
Reviewer 1 Report
This is an interesting study. Please clarify why did you chose nonparametrical Mann–Whitney U comparison tests and not parametrical ones.
Please also explicitly specify if your results confirm the two hypothesis (lines 81-89).
Moreso, in the discussion section try to add more studies then (6, 17, 18). Also please clarify the self citations and the situations of using data from a larger study.
Author Response
- This is an interesting study. Please clarify why did you chose nonparametrical Mann–Whitney U comparison tests and not parametrical ones.
Because of a large difference in sample size between the two groups (94 vs 1065 participants) and associated risk for assumption violation of parametric tests, non parametric Mann-Whiteny U comparisons test have been used.
We have thus added the following sentence in the section 2.3 Data analyses, on page 4, lines 168-171:
Non parametric test were used because of the large difference in sample size between the two groups and associated risk for assumption violation of parametric tests.
- Please also explicitly specify if your results confirm the two hypothesis (lines 81-89).
Following Reviewer’s 1 remark, we have added the following sentence in the section 4 Discussion, on page 6, lines 231-235:
In the binary regression analysis, only being a man and motives for using Tinder significantly predicted group status, thus corroborating our hypotheses. However, in contrast to our expectations, there was no significant effect of impulsivity-traits, nor significant interaction between motives for using Tinder and impulsivity-traits in predicting group membership.
- Moreso, in the discussion section try to add more studies then (6, 17, 18). Compléter les références
Following Reviewer 1’s suggestion, we have added the following references in the section 4 Discussion, on page 7, line 241, including a systematic review of psychosocial correlates of dating apps use:
- Timmermans, E., & De Caluwé, E. (2017). To Tinder or not to Tinder, that's the question: An individual differences perspective to Tinder use and motives. Personality and Individual Differences, 110, 74-79. https://doi.org/10.1016/j.paid.2017.01.026
- Sumter, S. R., & Vandenbosch, L. (2019). Dating gone mobile: Demographic and personality-based correlates of using smartphone-based dating applications among emerging adults. New Media & Society, 21(3), 655–673. https://doi.org/10.1177/1461444818804773
- Castro, Á., & Barrada, J. R. (2020). Dating apps and their sociodemographic and psychosocial correlates: A systematic review. International Journal of Environmental Research and Public Health, 17(18), 6500. https://doi.org/10.3390/ijerph17186500
- Also please clarify the self citations and the situations of using data from a larger study.
As requested by Reviewer 1, we have more specifically clarified the self-citation and the use of secondary data on page 4, lines 156-161. The paragraph reads as follows:
The study is secondary analysis of a previous study on Tinder use (12) and is part of a larger project on cybersex. The original study (12) focused on the various psychological profiles of Tinder users, whereas the current study more specifically aims to compare payers vs. non payers and determine which factors account for group membership. The same data set was thus used in the current study with the inclusion of an additional variable of interest «payers vs. non payers».
Reviewer 2 Report
This paper analise the psychological correlates of teh willingness to pay for a dating app, the Tinder app. This app have a free version and a pay app version for obtain more services. It´s a paper original and of interest.
This is a interesting study relate with the uso of apps, and specifically with the very utilized app, as is Tinder, along the word, with the objective to help to finding potencial romatic and/or sexual partners. Analize the potential of addiction of this type of app is relevant, as is relevant know other psychological characteristics of the users.
The study fave a good sample but utilize secondary data of a more large study. All the sample is heterosexual.
The results are relevant with clear differences between the pay and non pay groups in the utilization of app Tinder.
The authors need clarify o change several questions, as follow:
- The authors need a major support to the objective or hypothesis of the study in the introduction (ex., a more explanation to justify the hypothetized relation between impulsivity trait and more pay of Tinder; or the relation between depression and more Tinder app of pay).
- The sample need a more detailed description in relation to answer/non answer of the subjects and the recruitment procedure.
- The authors need indicate the fiability of the scales utilizaed in the sample of your study.
- In the table is indicate gender. ¿Really assess gender or sex? If assess gender, assess sex? Not it´s the same gender that sex.
- The authors need introduce as a limitation of the study that is remove 1/3 of the initial participants of the original sample.
- The section of conclussions need be rewrite because insist in the future studies and not in the central question of conclussions of the study. Or you need change the title of the section of conclussion.s.
Author Response
- The authors need a major support to the objective or hypothesis of the study in the introduction (ex., a more explanation to justify the hypothetized relation between impulsivity trait and more pay of Tinder; or the relation between depression and more Tinder app of pay).
As requested by Reviewer 2, we have provided more support for our hypothesis regarding impulsivity, depressive mood and paying for improving the experience of Tinder use on page 2, lines 84-91:
Indeed, as these two dimensions of impulsivity have been associated with poor inhibitory control and difficulties in overcoming immediate gratification (14), we might expect that the possibility to immediately improve the experience on Tinder by paying for extra-service is particularly appealing for impulsive users. As several studies showed that Tinder use has been associated with lower well-being (e.g., depressive mood, anxiety) (12, 15) and that depressive mood frequently co-occurs with impulsivity, especially the urgency trait (16), depressive mood has been used as a control variable in the current study.
Note that we have added the following references:
(14) Rochat, L., Billieux, J., Gagnon, J., & van der Linden, M. (2018). A multifactorial and integrative approach to impulsivity in neuropsychology : Insights from the UPPS model of impulsivity. Journal of Clinical and Experimental Neuropsychology, 40(1), 45‑61. https://doi.org/10.1080/13803395.2017.1313393
(15) Her, Y.C., & Timmermans, E. (2021) Tinder blue, mental flu? Exploring the associations between Tinder use and well-being. Information, Communication & Society, 24, 1303-1319. https://doi.org/10.1080/1369118X.2020.1764606
(16) Berg, J. M., R. D. Latzman, Bliwise, N. G., & Lilienfeld, S. O. (2015). Parsing the heterogeneity of impulsivity : A meta-analytic review of the behavioral implications of the UPPS for psychopathology. Psychological Assessment, 27(4), 1129‑1146. https://doi.org/10.1037/pas0000111
- The sample need a more detailed description in relation to answer/non answer of the subjects and the recruitment procedure.
Following Reviewer 2’s remark, we have given more details on the sample and recruitment procedure on page 2, lines 92-96. Note that missing or incomplete data regard 322 participants and not 538 as initially stated in the previous draft. Indeed, 216 participants have also been removed inasmuch as we focused on heterosexual participants only. The paragraph reads now as follows (pages 2-3):
Participants were recruited on various social networking sites (such as facebook, and,instagram) where a link to the survey was posted. They were all English-speaking Tinder users and were at least 18 years old. After removing 538 participants (322 with too many missing or incomplete data and 216 who reported non heterosexual orientation), the final sample consists of 1159 subjects (46.59% women).
- The authors need indicate the fiability of the scales utilizaed in the sample of your study.
As requested by Reviewer 2, we have now provided the internal consistency (Cronbach’s alpha) of each scale used in the current study in the Methods section (pages 3-4). Note that all Cronbach’s alpha fall in the acceptable range (> .70).
- In the table is indicate gender. ¿Really assess gender or sex? If assess gender, assess sex? Not it´s the same gender that sex.
According to Reviewer 2, we have removed the term “gender” and replace it by the term “sex” in Table1 (page 4), Table 2 (page 5) as well as in the Appendix A (page 8), which is more in line with the objectives of the study.
- The authors need introduce as a limitation of the study that is remove 1/3 of the initial participants of the original sample.
As described in the response to Reviewer 2’s comment above, 322 and not 528 subjects have been removed because of too many missing or incomplete data. Although 19 % of loss due to missing or incomplete data is not unusual in online survey study, we acknowledge that it may constitutes a limitation to the study. We have added the following sentence on page 7, lines 265-267:
Finally, the number of participants with too many missing or incomplete data (19% of the initial sample) who have been removed from the final sample may question the representativeness of the current sample.
- The section of conclusions need be rewrite because insist in the future studies and not in the central question of conclusions of the study. Or you need change the title of the section of conclusion.s.
According to Reviewer’s 2 remark, we have modified the title which now reads as follows (on page 7, line 268): Conclusion and further perspectives. We have also slightly modified the beginning of the paragraph (lines 269-273) which now reads as follows:
The current study adds to the field in terms of greater knowledge on psychosocial determinants of dating apps use and misuse. These results can have clinical utility, considering the needed personalized and multi-dimensional approach in assessment and psychotherapy of social networks’ misuse (28) as well as specifically those related to sexual behaviors (29).